# CD34^+^CD38^−^CD123^+^ Leukemic Stem Cell Frequency Predicts Outcome in Older Acute Myeloid Leukemia Patients Treated by Intensive Chemotherapy but Not Hypomethylating Agents

**DOI:** 10.3390/cancers12051174

**Published:** 2020-05-06

**Authors:** François Vergez, Marie-Laure Nicolau-Travers, Sarah Bertoli, Jean-Baptiste Rieu, Suzanne Tavitian, Pierre Bories, Isabelle Luquet, Véronique De Mas, Laetitia Largeaud, Audrey Sarry, Françoise Huguet, Eric Delabesse, Emilie Bérard, Christian Récher

**Affiliations:** 1Laboratoire d’Hématologie, Centre Hospitalier Universitaire de Toulouse, Institut Universitaire du Cancer de Toulouse Oncopole, 31059 Toulouse, France; nicolau-travers.marie-laure@iuct-oncopole.fr (M.-L.N.-T.); rieu.jb@chu-toulouse.fr (J.-B.R.); luquet.isabelle@iuct-oncopole.fr (I.L.); DeMas.Veronique@iuct-oncopole.fr (V.D.M.); largeaud.laetitia@iuct-oncopole.fr (L.L.); delabesse.eric@iuct-oncopole.fr (E.D.); 2Medicine Faculty, Université Toulouse III Paul Sabatier, 31330 Toulouse, France; bertoli.sarah@iuct-oncopole.fr; 3Cancer Research Center of Toulouse, UMR1037-INSERM, ERL5294 CNRS, 31100 Toulouse, France; 4Service d’Hématologie, Centre Hospitalier Universitaire de Toulouse, Institut Universitaire du Cancer de Toulouse Oncopole, 31059 Toulouse, France; tavitian.suzanne@iuct-oncopole.fr (S.T.); sarry.audrey@iuct-oncopole.fr (A.S.); huguet.francoise@iuct-oncopole.fr (F.H.); 5Réseau Onco-occitanie, Institut Universitaire du Cancer de Toulouse Oncopole, 31059 Toulouse, France; bories.pierre@iuct-oncopole.fr; 6Service d’Epidémiologie, Centre Hospitalier Universitaire de Toulouse, 31300 Toulouse, France; emilie.berard@univ-tlse3.fr

**Keywords:** leukemic stem cells, AML, chemoresistance, intensive chemotherapy, hypomethylating agents

## Abstract

The prognostic impact of immunophenotypic CD34^+^CD38^−^CD123^+^ leukemic stem cell (iLSC) frequency at diagnosis has been demonstrated in younger patients treated by intensive chemotherapy, however, this is less clear in older patients. Furthermore, the impact of iLSC in patients treated by hypomethylating agents is unknown. In this single-center study, we prospectively assessed the CD34^+^CD38^−^CD123^+^ iLSC frequency at diagnosis in acute myeloid leukemia (AML) patients aged 60 years or older. In a cohort of 444 patients, the median percentage of iLSC at diagnosis was 4.3%. Significant differences were found between treatment groups with a lower median in the intensive chemotherapy group (0.6%) compared to hypomethylating agents (8.0%) or supportive care (11.1%) (*p* <0.0001). In the intensive chemotherapy group, the median overall survival was 34.5 months in patients with iLSC ≤0.10% and 14.6 months in patients with >0.10% (*p* = 0.031). In the multivariate analyses of this group, iLSC frequency was significantly and independently associated with the incidence of relapse, event-free, relapse-free, and overall survival. However, iLSC frequency had no prognostic impact on patients treated by hypomethylating agents. Thus, the iLSC frequency at diagnosis is an independent prognostic factor in older acute myeloid patients treated by intensive chemotherapy but not hypomethylating agents.

## 1. Introduction

Acute myeloid leukemia (AML) arises from self-renewing leukemic stem cells (LSCs), which represent a tiny minority of all leukemic cells at diagnosis [1]. These LSCs are enriched within the CD34^+^CD38^−^ subpopulation, although not restricted to this phenotype [2,3]. It has long been shown that CD34^+^CD38^−^ AML cells are more resistant than the leukemic bulk population to chemotherapeutic agents of the AML backbone chemotherapy (i.e., anthracyclines and cytarabine) [4,5,6]. Moreover, a gene signature established from functionally-defined LSCs has an independent and strong prognosis impact on both initial responses to induction chemotherapy and overall survival, demonstrating the clinical relevance of the abundance of stem cells in AML [7]. This has been further strengthened by studies showing that patients who relapsed present an enriched population of LSCs [8] with increased LSC-related gene signatures [9].

We have shown that the percentage of immunophenotypic CD34^+^CD38^−^CD123^+^ leukemic stem cells (iLSCs) at diagnosis has a significant impact on complete response and survival in younger AML patients treated by intensive chemotherapy [10]. We used the CD123 marker because it enables the discrimination in the CD34^+^CD38^−^ compartment between normal hematopoietic stem cells that little express CD123 and leukemic stem cells that are positive for this marker [11]. Similar results have also been recently published using the CD34^+^CD38^−^ subpopulation as a surrogate for LSCs [12,13], although it is acknowledged that these phenotypes do not account for all functionally defined LSCs and may contain leukemic cells without stem cell properties. However, the role of iLSCs in older patients has not been fully assessed to date, although one study showed that a high peripheral blood CD34^+^CD38^−^ blast frequency was significantly associated with reduced complete response rates and poor overall survival [14]. In addition, the impact of iLSCs in patients treated by hypomethylating agents is currently unknown.

The goal of this study was to assess the impact of iLSCs frequency at diagnosis in older patients treated by intensive chemotherapy, hypomethylating agents, low-dose cytarabine, or best supportive care.

## 2. Results

### 2.1. Characteristics and Outcome of Patients According to Treatment Choice

In this cohort of 444 AML patients ≥60 years, 168 (37.8%) received intensive chemotherapy whereas 117 (26.4%), 16 (3.6%), and 143 (32.2%) were treated by hypomethylating agents, low-dose cytarabine and supportive care, respectively. Their characteristics are presented in Table 1. As expected, age, performance status, comorbidities, AML status (de novo versus secondary), white blood cell count, and cytogenetic risk were significantly different across these treatment subgroups. Biochemical parameters including serum albumin, ferritin, bilirubin, and LDH were also significantly different according to treatments. Furthermore, the distribution of mutations also varied significantly with *NPM1* and *IDH2*^R140^ mutations more frequently identified in patients treated by intensive chemotherapy. In the intensive chemotherapy group, 141 patients (84.0%) received the idarubicin-cytarabine-lomustine triplet, whereas 18 (10.7%) received idarubicin and 3 (1.8%) daunorubicin-cytarabine doublet. Thirty patients (17.9%) were allografted in the first complete response. In the hypomethylating agents’ group, most patients received azacitidine (94.9%), and the median number of cycles was 6 (interquartile range, IQR, 2–12). Only one patient was allografted in this group.

The response to treatment, early death rate, and outcome according to treatments are shown in Table 2. The median OS was 22.6 (6.2-not reached), 10.5 (4.1–19.6), 3.6 (2.4–7.0), 1.1 (0.5–3.5) with intensive chemotherapy, hypomethylating agents, low-dose cytarabine, and supportive care, respectively.

### 2.2. CD34^+^CD38^−^CD123^+^ Leukemic Stem Cells at Diagnosis

Multiparameter flow cytometry analyses were performed on 444 bone marrow samples (Appendix A). The median % of CD34^+^CD38^−^CD123^+^ at diagnosis was 4.3 (IQR, 0.3–20.2; minimum–maximum, 0.0–93.9). Significant differences in CD34^+^CD38^−^CD123^+^ % were found between treatment groups with a lower median in intensive chemotherapy group (0.6%, IQR, 0.1–8.7) compared to hypomethylating agents (8.0%, IQR, 1.1–29.9) or supportive care (11.1%, IQR, 0.8–36.9) (*p* <0.0001) (Table 3).

### 2.3. Prognostic Impact of CD34^+^CD38^−^CD123^+^ Leukemic Stem Cells in Patients Treated by Intensive Chemotherapy

The median follow-up in this group was 39.2 months (IQR, 29.1–55.2). Significant cut-offs of CD34^+^CD38^−^CD123^+^ % for the main endpoints were investigated by using the method of restricted cubic splines in multivariate analyses. The cut-off significantly and independently associated with OS was 0.10% (Appendix A). Median OS was 34.5 months (IQR, 12.2-not reached) in patients with CD34^+^CD38^−^CD123^+^ % ≤0.10% and 14.6 months (IQR, 6.6-not reached) in patients with >0.10% (*p* = 0.031) (Figure 1A). One-year (y), 3-y, and 5-y OS was 77.6% (95% confidence interval, CI, 65–86), 47.4% (95% CI, 33–61) and 43.5% (95% CI, 28–58) in patients with CD34^+^CD38^−^CD123^+^ % ≤0.10% and 54.5% (95% CI, 45–63), 32.9% (95% CI, 24–42) and 25% (95% CI, 16–35) in patients with >0.10%. In multivariate analysis, CD34^+^CD38^−^CD123^+^ % >0.10% was associated with a worse OS (HR: 1.86; 95% CI: 1.18–2.88, *p* = 0.007) after adjustment for white blood cell count, albumin and cytogenetic risk, which were all significantly associated with OS (Table 4).

The cut-off significantly and independently associated with EFS and CIR was 10% (Appendix A). Median EFS was 16.4 months (IQR, 5.6-not reached) in patients with CD34^+^CD38^−^CD123^+^ % ≤10% and 7.2 months (IQR, 3.1–23.3) in patients with >10% (*p* = 0.002) (Figure 1B). In multivariate analysis, CD34^+^CD38^−^CD123^+^ % >10% was associated with a worse EFS (HR: 2.33; 95% CI: 1.54–3.54, *p* <0.001) after adjustment for age, albumin, cytogenetic risk, and allogeneic stem-cell transplantation as a time-dependent variable, which were all significantly associated with EFS (Table 4). In the univariate Fine and Gray competing risks model, a CD34^+^CD38^−^CD123^+^ % >10% was associated with a significantly higher risk of relapse (HR: 1.77; 95% CI: 1.07–2.93, *p* = 0.026) (Figure 1C). In multivariate analysis, only CD34^+^CD38^−^CD123^+^ % >10% (HR: 2.0; 95% CI: 1.24–3.20, *p* = 0.004) and allogeneic stem-cell transplantation retained a significant impact on CIR (Table 4).

The cut-off significantly and independently associated with RFS was 0.10% (Appendix A). Median RFS was 31.8 months (IQR, 11.9-not reached) in patients with CD34^+^CD38^−^CD123^+^ % ≤ 0.10% and 16.1 months (IQR, 4.7-not reached) in patients with >0.10% (*p* = 0.032) (Figure 1D). In multivariate analysis, CD34^+^CD38^−^CD123^+^ % >0.10% was associated with a worse OS (HR: 1.75; 95% CI: 1.06–2.86, *p* = 0.027) after adjustment for albumin and allogeneic stem-cell transplantation, which were all significantly associated with RFS (Table 4).

Of note, the CD34^+^CD38^−^CD123^+^ % was not significantly and independently associated with day 30 (*P* = 0.700) and day 60 death (*p* = 0.124) as well as complete response (*p* = 0.191).

As there was no significant interaction between CD34^+^CD38^−^CD123^+^ % and age, cytogenetics, or ELN classification for each endpoint, we concluded that its impact on prognosis was not significantly different according to age as well as cytogenetic or molecular risk groups.

### 2.4. Characteristics of Older AML Patients Treated by Intensive Chemotherapy With a High Leukemic Stem Cell Burden

To better describe the characteristics of patients expressing higher CD34^+^CD38^−^CD123^+^ %, patients of the intensive chemotherapy group were split into 2 groups according to the significant cut-off values for OS and EFS (Table 5). A CD34^+^CD38^−^CD123^+^ % >0.10% was significantly associated with better performance status, secondary AML, lower white blood cell count, bone marrow blast %, and ferritin levels. No significant relationship between the % of CD34^+^CD38^−^CD123^+^ and *FLT3*-ITD, *IDH1*^R132^, *IDH2*^R140^, *IDH2*^R172^, and *DNMT3A* were identified whereas 33% of patients with CD34^+^CD38^−^CD123^+^ % >0.10% had *NPM1* mutations as compared to 64.6% of patients with ≤0.10% (*p* = 0.0003). There was also less *CEBPA* mutations in patients with CD34^+^CD38^−^CD123^+^ >0.10% although this difference was not statistically significant (3.1% versus 20.0% in patients with CD34^+^CD38^−^CD123^+^ ≤0.10%, *p* = 0.0656).

A CD34^+^CD38^−^CD123^+^ % >10% was significantly associated with better performance status, secondary AML, poor cytogenetic and ELN 2010 risk, higher albumin levels, and lower ferritin levels as compared to patients with ≤10%. Only 9.7% of patients had *NPM1* mutations as compared to 53.7% of patients with ≤10% (*p* < 0.0001).

### 2.5. Prognostic Impact of CD34^+^CD38^−^CD123^+^ Leukemic Stem Cells in Patients Treated by Hypomethylating Agents

The median follow-up in this group was 43.8 months (IQR, 26.9–54.5). There was no cut-off significantly and independently associated with response (*p* = 0.191) and OS (*p* = 0.477) meaning that CD34^+^CD38^−^CD123^+^ % had no prognostic impact in patients treated by hypomethylating agents. Median OS was 10.7 months (IQR, 4.7–20.8) in patients with CD34^+^CD38^−^CD123^+^ % ≤8% (median) and 10.2 months (IQR, 3.6–18.0) in patients with >8% (*p* = 0.439) (Figure 2). One-year, 3-y and 5-y OS was 45.8% (95% CI, 33–58), 12.7% (95% CI, 5–23) and 0% in patients with CD34^+^CD38^−^CD123^+^ % ≤8% and 37.2% (95% CI, 25–50), 10.7% (95% CI, 4–21) and 0% in patients with >8%. In multivariate analysis, high white blood cell count (HR: 1.82; 95% CI: 1.42–2.33; *p* < 0.0001), serum albumin >30g/L (HR: 0.40; 95% CI: 0.29–0.54; *p* < 0.0001) and adverse cytogenetic risk (HR: 2.89; 95% CI: 1.14–7.32; *p* < 0.0001) but not CD34^+^CD38^−^CD123^+^ >8% (HR: 1.13; 95% CI: 0.74–1.74, *p* = 0.559) were significantly associated with overall survival.

## 3. Discussion

In this study, we have confirmed the prognostic impact of the immunophenotypic leukemic stem cell frequency at diagnosis in older patients treated by intensive chemotherapy, completing previous studies in younger patients [10,12,13]. A recent study has reported a similar impact of leukemic stem cell frequency in older AML [14]. However, in this study, blood but not bone marrow CD34^+^CD38^−^ leukemic stem cell frequency was associated with a lower response rate and poorer overall survival. In our study, we showed that the frequency of iLSCs in bone marrow was also a significant and independent risk factor in older AML. Whether differences in chemotherapy regimen used in both studies and the addition of CD123 marker could have a role to explain this discrepancy is unclear. We did not assess the level of CD34^+^CD38^−^CD123^+^ in blood for comparison with bone marrow.

Similar to Khan et al., we showed that immunophenotypic leukemic stem cell frequency was correlated with a lower white blood cell count, an adverse cytogenetic risk, and less frequent *NPM1* mutations [14]. We also found other correlations with regards to general condition at diagnosis and biochemical markers, including serum albumin and ferritin levels. This could reflect cell proliferation status and hyperleukocytosis, often associated with poorer performance status at diagnosis, *NPM1* mutations, and altered levels of albumin and ferritin [15]. We can, therefore, identify two types of clinical presentations according to the frequency of leukemic stem cells: a slowly progressive disease with a preserved general condition, low white blood cell count, secondary AML, adverse cytogenetics, and a high frequency of iLSCs, as opposed to a more aggressive disease, poorer performance status, hyperleukocytosis, intermediate cytogenetic risk, *NPM1* mutations, and a low frequency of iLSCs.

It is unclear why the iLSC frequency had a significant and reproducible prognostic impact in patients treated by intensive chemotherapy but not hypomethylating agents. It has been shown that azacitidine fails to eradicate leukemic stem cells in patients with AML or myelodysplastic syndrome [16]. Indeed, expansion of two populations of leukemic stem cells, including lymphoid-primed multipotential progenitor-like and granulocyte-monocyte progenitor-like leukemic stem cells, was detected in patients treated by azacitidine even after achieving complete morphologic response. Moreover, it has been shown that the response to decitabine in chronic myelomonocytic leukemia is not associated with a decrease in the mutation allele burden, nor prevention of new genetic alteration occurrence [17]. Venetoclax plus azacitidine will likely become the new standard of care in the population of patients unfit for intensive chemotherapy. This combination is not yet approved in Europe and could not have been explored in our study. Anti-apoptotic BCL-2 is overexpressed in the AML stem cell population, and it has been shown that CD34^+^CD38^−^CD123^+^ cells were rapidly depleted in the bone marrow after this doublet therapy, indicating that adding venetoclax to azacitidine may affect the pool of LSCs contrary to azacitidine alone [18,19]. On the other hand, we and others have shown that iLSC frequency is associated with poor-risk cytogenetics, a subgroup of patients that do not do so well upon venetoclax-azacitidine treatment [20]. Thus, it remains to be determined if iLSC frequency has a prognostic impact in patients treated with venetoclax based-combinations.

It is noteworthy that some patients with a high CD34^+^CD38^−^CD123^+^ leukemic stem cell frequency have obtained a sustained response and a reasonably good overall survival, with approximately one-third of patients alive at three years. Although it was generally assumed that chemoresistance lies in leukemic stem cells because of their quiescence and immaturity, we have recently demonstrated by using limiting dilution assay in patient-derived xenograft models, that cytarabine does eradicate leukemic stem cells at least in some AML samples suggesting that leukemic stem cells are heterogeneous with regards to chemoresistance [21]. Therefore, the prognostic impact of iLSC frequency could be related to the relative proportion of chemoresistant leukemic stem cells. New specific chemoresistance markers will be needed to improve the prognostic power of leukemic stem cell frequency at diagnosis. Interestingly, cytarabine-resistant cells have mitochondrial-specific oxidative and bioenergetics features, suggesting that new markers related to this metabolism could improve the accuracy of leukemic stem cell phenotyping in order to better predict chemoresistance and outcome in AML patients treated by cytotoxic therapies [21,22].

The study has several limitations. The observational nature of the study does not allow for a possible confusion bias to be perfectly controlled, although analyses were adjusted to the major confounding factors expected in AML. Moreover, it’s not clear why the predictive cut-off was 0.10% for OS and RFS, on the one hand, and 10% for EFS and CIR, on the other hand. This seems to indicate a better sensitivity of the iLSC to predict death than relapse, but this aspect has to be confirmed. Finally, there could also be a lack of power for the hypomethylating agent population (even if the results are frankly not statistically significant), stressing the need for additional and prospective studies to confirm our results.

## 4. Materials and Methods

### 4.1. Patients

The inclusion criteria were: newly diagnosed AML according to the WHO 2008 classification and age 60 years or older. Patients with acute promyelocytic leukemia were excluded. This study included 444 patients admitted at the Hematology department of Toulouse University Hospital-IUCT-O and/or registered in the regional oncology network from 1 January 2011 to 31 December 2017. Data were gathered on an electronic clinical research form. Written informed consent was obtained from all patients in accordance with the Declaration of Helsinki, allowing the collection of clinical and biological data in an anonymized database (CRE IUCT-O: 2-19-04). Cytogenetic and molecular risk classifications were in accordance with the Medical Research Council and ELN 2010 classifications, respectively [23,24]. Details on treatment with a hypomethylating agent and first-line chemotherapy regimen used over time have been reported elsewhere [25,26,27]. The decision-making process with regard to intensive or non-intensive treatment was based on initial characteristics such as white blood cell count (WBC), cytogenetics, age, secondary AML, performance status, and comorbidities. Briefly, the first issue was to judge if patients could benefit from intensive chemotherapy. If not, the second issue was to determine if patients could benefit from a hypomethylating agent, and if not, those patients were referred to low dose cytarabine or supportive care [25].

### 4.2. Assessment of Efficacy

Endpoints, including response, event-free survival (EFS), the cumulative incidence of relapse (CIR), relapse-free survival (RFS), and overall survival (OS), were assessed according to standard criteria [28]. Deaths at day 30 and day 60 were calculated from day 1 of treatment or from the date of diagnosis in patients receiving supportive care only. Bone marrow assessment in patients treated with intensive chemotherapy was performed after blood recovery or in case of delayed recovery between days 35 and 45. In the hypomethylating agent and low-dose cytarabine groups, bone marrow aspiration was carried out after 3 to 6 cycles of treatment.

### 4.3. Analysis of Leukemic Stem Cells

All AML samples were obtained from the bone marrow at diagnosis. CD45 staining and side scatter (SS) properties were used to isolate the leukemic cell populations, referred to as the bulk of the leukemia and usually defined by weak CD45 expression (CD45 dim) and low SS (SSlow). For AML samples with monocytic differentiation in which blast cells can be found in the monocytic gate, the gating strategy was made in agreement with the morphological study. The percentage (%) of CD34^+^CD38^−^CD123^+^ cells was then quantified as the ratio between the numbers of CD34^+^CD38^−^CD123^+^ cells and CD45 dim/SSlow cells [10] (Appendix A). Analyses were performed in the routine setting at the time of diagnosis work-up without any information regarding the treatment choice. Assessment of AML minimal residual disease by flow cytometric methods was not available during the study period.

### 4.4. Statistical Analyses

Before doing any analyses, we assessed the power of the study to assess the impact in OS of iLSCs frequency at diagnosis in older patients treated by intensive chemotherapy, hypomethylating agents, low-dose cytarabine, or best supportive care: 350 deaths provided a power greater than 80% to detect a Hazard Ratio (HR) for OS >1.40 (for high level vs. low level of iLSCs) with a two-sided type-1 error rate of 5% (alpha = 0.05) for the comparison of two exponential survival distributions [29].

We first described patients’ characteristics using number and frequency for qualitative data; median and interquartile range (IQR) for quantitative data. Differences in early death and response rate were tested in univariate analyses using Chi^2^ test (or Fisher’s exact test in case of small expected numbers). Multivariate analyses of early death and response rate were conducted using logistic regression. For univariate survival analyses of OS, EFS, and RFS, Kaplan–Meier survival curves were drawn, and differences in survival functions were tested using the Log-Rank test. Univariate survival analyses used cumulative incidence functions and Gray’s test for relapse (CIR) since non-relapse mortality was treated as competing events. Hazard ratios (HR) and 95% confidence intervals (CI) were assessed using a standard Cox model for OS, EFS, and RFS, and a proportional subdistribution hazard model which is an extension of the Cox model for the situation of competing risks, for CIR [30]. Multivariate analyses initially included CD34^+^CD38^−^CD123^+^ % levels, together with potential confounding factors associated with endpoints (*p* <0.20) in univariate analysis. Age, sex, ECOG performance status, Charlson comorbidity index, AML subtype, white blood cell count, albumin, serum ferritin, bone marrow blasts, and cytogenetic risk at diagnosis together with treatment (intensive chemotherapy, hypomethylating agent, low-dose cytarabine, or supportive care), and allogeneic stem-cell transplantation (only for patients treated with intensive chemotherapy) were assessed as potential confounding factors. Then we used a stepwise regression to assess variables that were significantly and independently associated with endpoints (*p* <0.05). The proportional-hazard assumption was tested for each covariate of the Cox model by the ‘log-log’ plot method curves and was always met. When the linearity hypothesis was not respected, continuous potential confounding factors were transformed into ordered data. Cut-offs of CD34^+^CD38^−^CD123^+^ % were chosen according to restricted cubic splines (RCS) method in multivariate analyses [31,32,33]. Using the RCS method, we obtained a continuous smooth function that is linear before the first knot, a piecewise cubic polynomial between adjacent knots and linear again after the last knot. The locations of the knots are determined by the percentiles recommended by Harrell et al. [34] (10th, 50th and 90th). Graphs showing the adjusted impact of CD34^+^CD38^−^CD123^+^ % according to RCS method were done. Interactions between CD34^+^CD38^−^CD123^+^ % and independent factors were tested in final models. Allogeneic stem-cell transplantation was evaluated as a time-dependent covariate. All reported *p*-values were two-sided and the significance threshold was <0.05. Statistical analyses were performed on STATA® version 14.1 (STATA Corp., College Station, TX, USA).

## 5. Conclusions

Our results showed that the independent prognostic significance of high levels of CD34^+^CD38^−^CD123^+^ leukemic cells at diagnosis is also observed in older AML patients treated by intensive chemotherapy and could be used to stratify risk in those patients. However, this prognostic marker should not be used in patients treated by hypomethylating agents.

## Figures and Tables

**Figure 1 cancers-12-01174-f001:**
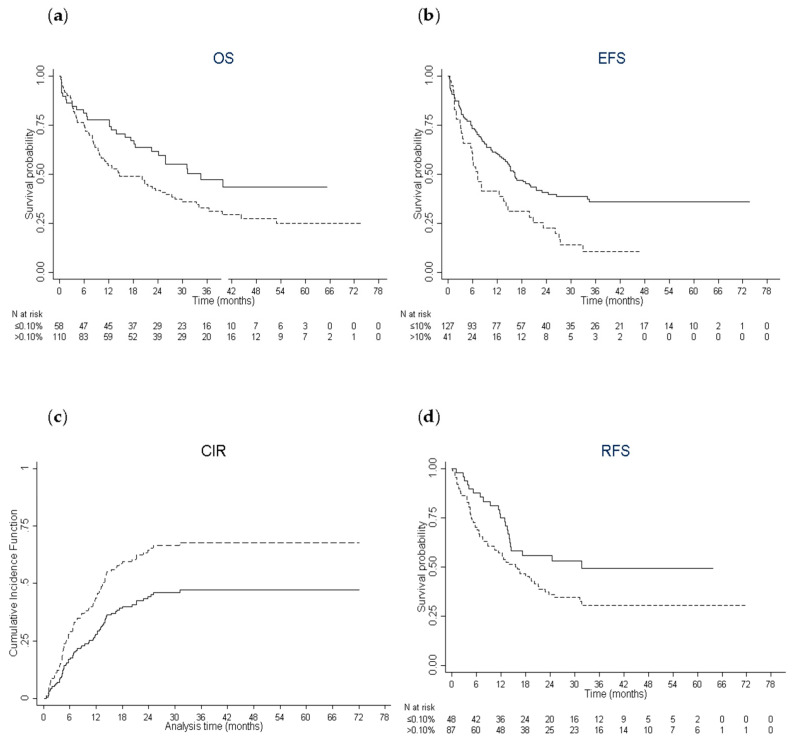
Estimates of survival end points and incidence of relapse. (**a**) Kaplan–Meier curves for overall survival according to the leukemic stem frequency (CD34^+^CD38^−^CD123^+^ % ≤0.10%: solid line; >0.10%: dashed line) in patients treated by intensive chemotherapy. (**b**) Kaplan–Meier curves for event-free survival (CD34^+^CD38^−^CD123^+^ % ≤10%: solid line; >10%: dashed line). (**c**) Cumulative incidence of relapse (CD34^+^CD38^−^CD123^+^ % ≤10%: solid line; >10%: dashed line). (**d**) Kaplan–Meier curves for relapse-free survival (CD34^+^CD38^−^CD123^+^ % ≤0.10%: solid line; >0.10%: dashed line).

**Figure 2 cancers-12-01174-f002:**
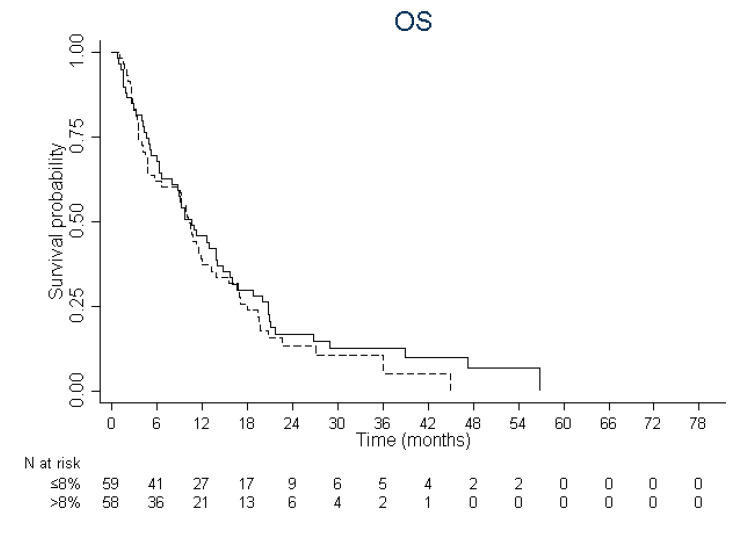
Kaplan–Meier curves for overall survival according to the leukemic stem frequency in patients treated by hypomethylating agents (CD34^+^CD38^−^CD123^+^ % ≤8%: solid line; >8%: dashed line).

**Table 1 cancers-12-01174-t001:** Characteristics of AML patients ≥60 years according to treatment choice.

Characteristics	Intensive Chemo *N* = 168	HMA *N* = 117	LDAC *N* = 16	Supportive Care *N* = 143	*p*-Value
**Sex, *n*. (%)**					0.5142
Male	96 (57.1)	66 (56.4)	9 (56.3)	92 (64.3)	
Female	72 (42.9)	51 (43.6)	7 (43.8)	51 (35.7)	
**Age, years**					
Median (IQR)	66.9 (63.6–71.8)	76.5 (72.0–81.8)	76.7 (72.7–80.2)	79.5 (73.4–84.6)	<0.0001
≤70, *n.* (%)	113 (67.3)	21 (17.9)	3 (18.8)	19 (13.3)	
70–80, *n*. (%)	55 (32.7)	57 (48.7)	9 (56.3)	58 (40.6)	<0.0001
>80, *n*. (%)	0 (0.0)	39 (33.3)	4 (25.0)	66 (46.2)	
**CCI ≥1, *n*. (%)**	50 (29.9)	47 (41.2)	8 (50.0)	74 (64.3)	<0.0001
**ECOG PS, n. (%)**					0.0004
0–1	115 (70.6)	71 (67.6)	8 (53.3)	47 (46.1)	
2–4	48 (29.4)	34 (32.4)	7 (46.7)	55 (53.9)	
**EMD, *n*. (%)**					0.0007
No	117 (69.6)	99 (86.8)	8 (50.0)	82 (77.4)	
Yes	51 (30.4)	15 (13.2)	8 (50.0)	24 (22.6)	
**AML subtype, *n*. (%)**					<0.0001
De novo AML	124 (73.8)	52 (44.4)	6 (37.5)	65 (48.9)	
Secondary	44 (26.2)	65 (55.6)	10 (62.5)	68 (51.1)	
**FAB, *n*. (%)**					<0.0010
M0	7 (4.2)	6 (5.5)	0 (0.0)	10 (8.5)	
M1	41 (24.6)	12 (11.0)	4 (26.7)	18 (15.3)	
M2	57 (34.1)	60 (55.1)	6 (40.0)	40 (33.9)	
M4	29 (17.4)	5 (4.6)	2 (13.3)	19 (16.1)	
M5	15 (9.0)	4 (3.7)	2 (13.3)	8 (6.8)	
M6	4 (2.4)	13 (11.9)	0 (0.0)	5 (4.2)	
Unclassified	14 (8.4)	9 (8.3)	1 (6.7)	18 (15.3)	
**WBC, G/L**					
Median (IQR)	11.6 (2.7–60.8)	2.7 (1.6–8.5)	50.0 (7.0–95.8)	10.9 (2.5–35.7)	<0.0001
**Platelet count, G/L**					
Median (IQR)	64 (39–110)	73 (39–116)	48 (23–69)	53 (26–98)	0.0117
**BM blasts, %**					
Median (IQR)	57 (35–82)	32 (25–52)	43 (22–92)	43 (30–71)	<0.0001
≤30, n. (%)	29 (17.5)	52 (46.0)	5 (33.3)	37 (30.3)	<0.0001
>30, n. (%)	137 (82.5)	61 (54.0)	10 (66.7)	85 (69.7)	
**MLD, *n*. (%)**					0.0039
Yes	20 (12.3)	32 (29.6)	2 (14.3)	19 (17.3)	
No	143 (87.7)	76 (70.4)	12 (85.7)	91 (82.7)	
**Cytogenetics, *n*. (%)**					<0.0001
Favorable	8 (4.8)	0 (0.0)	0 (0.0)	1 (0.8)	
Intermediate	130 (77.4)	59 (51.3)	8 (50.0)	59 (46.8)	
Adverse	30 (17.9)	56 (48.7)	8 (50.0)	66 (52.4)	
**ELN 2010, *n*. (%)**					<0.0001
Favorable	41 (25.0)	3 (2.9)	0 (0.0)	7 (6.4)	
Intermediate-1	53 (32.3)	19 (18.6)	5 (33.3)	14 (12.8)	
Intermediate-2	40 (24.4)	24 (23.5)	2 (13.3)	22 (20.2)	
Adverse	30 (18.3)	56 (54.9)	8 (53.3)	66 (60.6)	
***FLT3*-ITD, *n*.(%)**					0.0579
Yes	34 (24.5)	5 (9.1)	5 (35.7)	10 (21.3)	
No	105 (75.5)	50 (90.9)	9 (64.3)	37 (78.7)	
***NPM1*, *n*. (%)**					<0.0001
Yes	61 (43.9)	6 (10.9)	3 (23.1)	10 (21.3)	
No	78 (56.1)	49 (89.1)	10 (76.9)	37 (78.7)	
***CEBPA*, *n*. (%)**				
Yes	5 (9.6)	1 (16.7)	0 (0.0)	0 (0.0)	0.6632
No	47 (90.4)	5 (83.3)	6 (100.0)	5 (100.0)	
***IDH1*^R132^, *n*. (%)**					0.1334
Yes	10 (12.5)	1 (2.5)	0 (0.0)	1 (2.9)	
No	70 (87.5)	39 (97.5)	3 (100.0)	34 (97.1)	
***IDH2*^R140^, *n*. (%)**					0.0317
Yes	15 (18.8)	2 (5.0)	0 (0.0)	1 (2.9)	
No	65 (81.3)	38 (95.0)	3 (100.0)	34 (97.1)	
***IDH2*^R172^, *n*. (%)**					0.9684
Yes	3 (3.8)	1 (2.6)	0 (0.0)	1 (2.9)	
No	76 (96.2)	38 (97.4)	3 (100.0)	34 (97.1)	
***DNMT3A***					0.5094
Yes	5 (16.7)	0 (0.0)	0 (0.0)	0 (0.0)	
No	25 (83.3)	3 (100.0)	0 (0.0)	4 (100.0)	
**Albumin, g/L**					
Median (IQR)	36.0 (31.5–40.0)	37.0 (34.0–40.0)	31.5 (30.0–39.0)	34.0 (27.0–38.0)	<0.0001
≥30 g/L, n. (%)	103 (62.8)	73 (71.6)	5 (41.7)	37 (42.5)	0.0002
<30 g/L, n. (%)	61 (37.2)	29 (28.4)	7 (58.3)	50 (57.5)	
**LDH, UI/L**				
Median (IQR)	499 (287–803)	407 (214–619)	622 (316–1076)	643 (305–1239)	0.0001
<Normal, n. (%)	39 (23.4)	42 (39.3)	4 (25.0)	20 (23.0)	0.0213
>Normal, n. (%)	128 (76.6)	65 (60.7)	12 (75.0)	67 (77.0)	
**Creatinine, µmol/L**					
Median (IQR)	81.0 (69.0–102.5)	82.5 (65.5–106.0)	87.5 (67.5–103.0)	92.0 (71.0–119.0)	0.1259
**Bilirubin, µmol/L**					
Median (IQR)	8.3 (6.0–10.3)	9.0 (6.8–14.0)	7.8 (5.4–11.5)	10.0 (6.9–15.0)	0.0040
**Fibrinogen, g/L**					
Median (IQR)	3.9 (3.1–4.8)	3.9 (3.1–5.0)	4.0 (2.4–5.8)	3.7 (3.1–5.0)	0.9461
**Serum ferritin, µg/L**					
Median (IQR)	713 (372–1321)	523 (263–1086)	1007 (651–3468)	996 (637–2109)	0.0010
≤900, n. (%)	97 (59.5)	49 (68.1)	3 (42.9)	28 (47.5)	0.0917
>900, n. (%)	66 (40.5)	23 (31.9)	4 (57.1)	31 (52.5)	

HMA, hypomethylating agents; LDAC, low dose cytarabine; IQR, interquartile range; CCI, modified Charlson comorbidity index; ECOG: Eastern cooperative oncology group; PS, performance status; EMD, extramedullary disease; FAB: French American British classification; WBC, white blood cell count; G/L, giga per liter; g/L: gram per liter; BM, bone marrow; MLD, multilineage dysplasia; ELN, European leukemia net.

**Table 2 cancers-12-01174-t002:** Responses and outcomes according to first-line treatments.

Clinical Parameters	Intensive Chemo *N* = 168	HMA *N* = 117	LDAC *N* = 16	Supportive Care *N* = 143
**CR, *n*. (%)**	135 (80.4)	21 (17.9)	0 (0.0)	NA
**Day-30 death, *n*. (%)**	17 (10.1)	6 (5.1)	2 (12.5)	58 (40.6)
**Day-60 death, *n*. (%)**	21 (12.5)	16 (13.7)	4 (25.0)	81 (56.6)
**EFS**		NA	NA	NA
Median, months (IQR)	14.6 (4.1–NR)			
1-y, % (95% CI)	56.0 (48.1–63.1)			
3-y	29.6 (22.4–37.1)			
5-y	29.6 (22.4–37.1)			
**RFS**		NA	NA	NA
Median, months (IQR)	17.3 (6.7–NR)			
1-y, % (95% CI)	63.6 (54.8–71.1)			
3-y	37.2 (28.4–45.9)			
5-y	37.2 (28.4–45.9)			
**CIR**				
1-y, % (95% CI)	31.9 (24.3–40.0)			
3-y	51.8 (42.8–60.4)			
5-y	51.8 (42.8–60.4)			
**OS**				
Median, months (IQR)	22.6 (6.2–NR)	10.5 (4.1–19.6)	3.6 (2.4–7.0)	1.1 (0.5–3.5)
1-y, % (95% CI)	62.5 (54.7–69.3)	41.6 (32.6–50.3)	12.5 (2.1–32.8)	7.7 (3.8–13.3)
3-y	37.9 (30.0–45.8)	11.8 (6.4–18.9)	NR	0.9 (0.1–4.2)
5-y	31.1 (22.7–39.8)	NR	NR	NR

HMA, hypomethylating agents; LDAC, low-dose cytarabine; CR: complete response; EFS; event-free survival; RFS, relapse-free survival; OS, overall survival; CIR: cumulative incidence of relapse; NR: not reached; NA: not applicable; IQR, interquartile range; CI, confidence interval.

**Table 3 cancers-12-01174-t003:** Distribution of the % of leukemic stem cells according to treatment choice.

LSCs (%)	Intensive Chemo *N* = 168	HMA *N* = 117	LDAC *N* = 16	Supportive Care *N* = 143	*p*-Value	All Patients *N* = 444
**Mean (SD)**	8.0 (15.7)	18.0 (21.9)	9.7 (14.2)	20.3 (23.8)	<0.0001	14.7 (20.9)
**Median**	0.6	8.0	2.3	11.1		4.3
**IQR**	0.1–8.7	1.1–29.9	0.4–16.9	0.8–36.9		0.3–20.2
**Min, Max**	0.0–93.9	0.0–85.3	0.0–52.7	0.0–92.2		0.0–93.9

LSC, leukemic stem cells; SD, standard deviation; IQR, interquartile range; HMA, hypomethylating agents; LDAC, low dose cytarabine; Min, minimum; Max, maximum.

**Table 4 cancers-12-01174-t004:** Multivariate analyses for OS, EFS, CIR, and RFS in patients treated by intensive chemotherapy.

Variable	HR	95% CI	*p*-Value
**Overall Survival**			
**LSC >0.10%**	1.85	1.18–2.88	0.007
**WBC >median**	1.84	1.44–2.37	<0.001
**Albumin ≥30 g/L**	0.44	0.32–0.61	<0.001
**Cytogenetic risk**			
Intermediate	2.13	0.86–5.27	0.102
Adverse	3.14	1.24–7.93	0.015
**Event free survival**			
**LSC >10%**	2.33	1.54–3.54	<0.001
**Age 70–80y**	1.62	1.08–2.44	0.019
**Albumin ≥30 G/L**	0.37	0.23–0.59	<0.001
**Cytogenetic risk**			
Intermediate	2.06	0.75–5.66	0.158
Adverse	4.51	1.50–13.57	0.007
**Allo-SCT**	0.47	0.24–0.92	0.027
**Cumulative incidence of relapse**			
**LSC >10%**	2.00	1.24–3.20	0.004
**Allo-SCT**	0.27	0.12–0.64	0.003
**Relapse free survival**			
**LSC >0.1%**	1.75	1.06–2.86	0.027
**Albumin ≥30 g/L**	0.49	0.28–0.86	0.013
**Allo-SCT**	0.51	0.27–0.97	0.039

HR, hazard ratio; CI confidence interval; LSC, leukemic stem cells; WBC, white blood cell count; G/L, gram per liter; Allo-SCT, allogeneic stem-cell transplantation (as a time-dependent variable).

**Table 5 cancers-12-01174-t005:** Characteristics of AML patients treated by intensive chemotherapy according to the % of leukemic stem cells (LSCs).

LSCs	≤0.10% *N* = 58	>0.10% *N* = 110	*p*-Value	≤10% *N* = 127	>10% *N* = 41	*p*-Value
**Sex, *n*. (%)**			0.7078			0.3506
Male	32 (55.2)	64 (58.2)		70 (55.1)	26 (63.4)	
Female	26 (44.8)	46 (41.8)		57 (44.9)	15 (36.6)	
**Age, years**						
Median (IQR)	66.9 (63.4–73.4)	66.9 (63.8–71.6)	0.4216	66.8 (63.3–72.0)	67.4 (65.2–70.9)	0.7761
≤70, *n*. (%)	39 (67.2)	74 (67.3)	1.0000	84 (66.1)	29 (70.7)	0.7026
70–80, *n*. (%)	19 (32.8)	36 (32.7)		43 (33.9)	12 (29.3)	
**CCI ≥ 1, *n*. (%)**	13 (22.4)	37 (33.9)	0.1213	40 (31.5)	10 (25.0)	0.4340
**ECOG PS, *n*. (%)**			0.0251			0.0271
0–1	34 (59.6)	81 (76.4)		82 (66.1)	33 (84.6)	
2–4	23 (40.4)	25 (23.6)		42 (33.9)	6 (15.4)	
**EMD, *n*. (%)**			0.6230			0.8615
Yes	19 (32.8)	32 (29.1)		39 (30.7)	12 (29.3)	
No	39 (67.2)	78 (70.9)		88 (69.3)	29 (70.7)	
**AML subtype, *n*. (%)**			0.0079			0.0315
De novo AML	50 (86.2)	74 (67.3)		99 (78.0)	25 (61.0)	
Secondary	8 (13.8)	36 (32.7)		28 (22.0)	16 (39.0)	
**WBC, G/L** Median (IQR)	26.4 (5.0–92.4)	7.1 (2.4–42.3)	0.0046	14.5 (2.8–68.8)	9.7 (2.5–46.0)	0.2561
**Platelet count, G/L** Median (IQR)	56 (29–102)	65 (39–117)	0.1261	68 (41–110)	54 (35–95)	0.1788
**BM blasts, %**						
Median (IQR)	80 (49–91)	48 (33–68)	<0.0001	57 (36–84)	55 (35–75)	0.4144
≤30, *n*. (%)	6 (10.3)	23 (21.3)	0.0764	22 (17.5)	7 (17.5)	0.9954
>30, *n*. (%)	52 (89.7)	85 (78.7)		104 (82.5)	33 (82.5)	
**MLD, *n*. (%)**			0.0143			0.0996
Yes	2 (3.6)	18 (16.8)		12 (9.8)	8 (20.0)	
No	54 (96.4)	89 (83.2)		111 (90.2)	32 (20.0)	
**Cytogenetics, *n*. (%)**			0.2400			0.0118
Favorable	1 (1.7)	7 (6.4)		5 (3.9)	3 (7.3)	
Intermediate	49 (84.5)	81 (73.6)		105 (82.7)	25 (61.0)	
Adverse	8 (13.8)	22 (20.0)		17 (13.4)	13 (31.7)	
**ELN 2010, *n*. (%)**			0.2752			0.0109
Favorable	19 (33.9)	22 (20.4)		37 (30.1)	4 (9.8)	
Intermediate-1	16 (28.6)	37 (34.3)		41 (33.3)	12 (29.3)	
Intermediate-2	13 (23.2)	27 (25.0)		28 (22.8)	12 (29.3)	
Adverse	8 (14.3)	22 (20.4)		17 (13.8)	13 (31.7)	
***FLT3*-ITD, *n*. (%)**			0.1762			0.7823
Yes	15 (31.3)	19 (20.9)		27 (25.0)	7 (22.6)	
No	33 (68.8)	72 (79.1)		81 (75.0)	24 (77.4)	
***NPM1*, *n*. (%)**			0.0003			<0.0001
Yes	31 (64.6)	30 (33.0)		58 (53.7)	3 (9.7)	
No	17 (35.4)	61 (67.0)		50 (46.3)	28 (90.3)	
***CEBPA*, *n*. (%)**			0.0656			1.0000
Yes	4 (20.0)	1 (3.1)		4 (9.3)	1 (11.1)	
No	16 (80.0)	31 (96.9)		39 (90.7)	8 (88.9)	
***IDH1*^R132^, *n*. (%)**			1.0000			0.4097
Yes	4 (12.9)	6 (12.2)		7 (10.9)	3 (18.8)	
No	27 (87.1)	43 (87.8)		57 (89.1)	13 (81.3)	
***IDH2*^R140^, *n*. (%)**			0.1983			0.7228
Yes	8 (25.8)	7 (14.3)		13 (20.3)	2 (12.5)	
No	23 (74.2)	42 (85.7)		51 (79.7)	14 (87.5)	
***IDH2*^R172^, *n*. (%)**			1.0000			1.0000
Yes	1 (3.3)	2 (4.1)		3 (4.8)	0 (0.0)	
No	29 (96.7)	47 (95.9)		60 (95.2)	16 (100.0)	
***DNMT3A***			1.0000			0.5561
Yes	2 (16.7)	3 (16.7)		5 (20.0)	0 (0.0)	
No	10 (83.3)	15 (83.3)		20 (80.0)	5 (100.0)	
**Albumin, g/L**						
Median (IQR)	35.5 (30.5–40.0)	36.0 (32.2–40.0)	0.2653	35.0 (30.0–39.0)	38 (35.0–40.5)	0.0203
≥30 g/L, *n*. (%)	32 (57.1)	71 (65.7)	0.2800	73 (58.4)	30 (76.9)	0.0366
<30 g/L, *n*. (%)	24 (42.9)	37 (34.3)		52 (41.6)	9 (23.1)	
**LDH, UI/L**						
Median (IQR)	448 (270–836)	530 (330–781)	0.9086	537 (295–836)	453 (241–676)	0.1218
<Normal, *n*. (%)	10 (17.5)	29 (26.4)	0.2014	27 (21.4)	12 (29.3)	0.3027
>Normal, *n*. (%)	47 (82.5)	81 (73.6)		99 (78.6)	29 (70.7)	
**Creatinine, µmol/L**						
Median (IQR)	85 (69–108)	80 (69–100)	0.4918	83 (68–105)	78 (74–91)	0.7507
**Bilirubin, µmol/L**						
Median (IQR)	7.6 (6.0–9.5)	8.4 (6.1–10.5)	0.4915	8.3 (6.1–10.0)	8.3 (6.0–11.0)	0.8970
**Fibrinogen, g/L**						
Median (IQR)	3.7 (2.9–4.7)	4.0 (3.3–4.9)	0.1069	4.0 (3.0–4.9)	3.9 (3.4–4.4)	0.8074
**Serum ferritin, µg/L**						
Median (IQR)	956 (554–1809)	578 (344–1199)	0.0039	761 (376–1477)	569 (370–862)	0.0588
≤900, *n*. (%)	26 (46.4)	71 (66.4)	0.0138	66 (53.7)	31 (77.5)	0.0076
>900, *n*. (%)	30 (53.6)	36 (33.6)		57 (46.3)	9 (22.5)	

IQR, interquartile range; CCI, modified Charlson comorbidity index; ECOG: Eastern cooperative oncology group; PS, performance status; EMD, extramedullary disease; WBC, white blood cell count; G/L, giga per liter; g/L: gram per liter; BM, bone marrow; MLD, multilineage dysplasia; ELN, European leukemia net.

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
