# Peer review of "CD34+CD38−CD123+ Leukemic Stem Cell Frequency Predicts Outcome in Older Acute Myeloid Leukemia Patients Treated by Intensive Chemotherapy but Not Hypomethylating Agents"

_cancers, 2020, doi:10.3390/cancers12051174_

Round 1
Reviewer 1 Report
The manuscript very concise and well written shows the results of a single study of CD123 expression and its correlation with disease and prognosis in old patients with acute myeloid leukemia. However, the article is not so novel but includes new data regard bone marrow of elderly patients. The study has many limitations due to the heterogeneity in terms of treatment which are described by the authors.
Major Comments:
- About 20% of AML cases are characterized by absence of neoplastic CD34+ cells. In these cases commonly small CD34+ (<1%) blast population does not contain Leukemia Stem Cells. By definition, these CD34- patients lack in CD34+CD38- and CD34+CD38+ leukemic populations.
How do the authors comment this sentence in relation to their discovery? How this model can analyze this type of patients?
- Where is a table indicating the cytogenetic characteristic of the patients. Are all these AML FLT3/ITD mutation.
Minor Comments:
- Add AML abbreviation in line 24.
- Remove repetitions in line 319
Author Response
We thank the Reviewer for the comments.
Major Comments:
- About 20% of AML cases are characterized by absence of neoplastic CD34+ cells. In these cases commonly small CD34+ (<1%) blast population does not contain Leukemia Stem Cells. By definition, these CD34- patients lack in CD34+CD38- and CD34+CD38+ leukemic populations.
1.How do the authors comment this sentence in relation to their discovery? How this model can analyze this type of patients?
Response:
Since 2013, 1830 AML samples have been phenotyped in our lab. The mean expression of CD34 in blasts cells is 60%. In this cohort, 279 AML (15.2%) have a very low of expression of CD34 (1% or less of expression of CD34). The leukemic status of these cells is still controversial: Schuurhuis et al. (PLOS ONE 2013) did not find AML mutations in CD34+ cells in CD34- AML, whereas Martelli et al. (Blood 2010) showed that LSC of NPM1+AML (usually CD34low) expressed CD34 and Quek et al. (J Exp Med 2016) found LSCs in CD34+ and CD34- compartments of CD34- AML.
In our lab, we analyze more events of CD34- AML (more than 1 000 000 events) allowing us to lower our threshold of detection of CD34+ cells to 0.1-0.01%. And CD34+CD38-CD123+ subpopulations are therefore detected in 64.6% of CD34- AML. Leukemic origin of these cells is established by expression of CD123. Furthermore, we found that in 35/135 CD34- AML (25.9%), CD34+CD38- subpopulations have a LMPP-like phenotype, in line with their leukemic origin (Goardon et al. Cancer Cell 2011).
To summary our data, in almost 95% of AML cases (CD34- and CD34+ AML), we are able to quantify a population of CD34+CD38- cells of 0.01% or greater. Especially in CD34- AML, we cannot assess their leukemic origin if not by their expression of CD123 and their LMPP-like phenotype (25% of CD34- AML).
So, in our model, we did not classify AML according to the expression of CD34. Indeed, in our cohort, 75 CD34- AML were included. 54 were treated by intensive chemotherapy including 24% with more than 0.1% of CD34+CD38-CD123+. Median OS of CD34- AML <0.1% CD34+CD38-CD123+ was 34.4 versus 28.1 months for the group of CD34- AML >0.1%.
Nevertheless, new LSC markers (TIM3, CD36 or others) will certainly be needed to replace CD34 and to study specifically the impact of LSC frequency in CD34- AML.
Major Comments:
2.Where is a table indicating the cytogenetic characteristic of the patients. Are all these AML FLT3/ITD mutation.
Response:
Cytogenetics and mutational status of patients are available in Table 1 (according to treatment) and in Table 5 (according to CD34+CD38-CD123+ groups).
Minor comments have been modified in the text.
Reviewer 2 Report
The manuscript entitled “Leukemic stem cell frequency predicts outcome in older acute myeloid leukemia patients treated by intensive chemotherapy but not hypomethylating agents” by Vergez et al. reveals the prognostic impact of the frequency of the CD34+CD38-CD123+ population in the bone marrow of patients with AML at the time of diagnosis. They apply this analysis to patients aged 60 years or older treated by standard induction therapy, hypomethylating agents or supportive care. They found that the frequency of the CD34+CD38-CD123+ population in the bone marrow of patients with AML at the time of diagnosis is an independent prognostic factor in older acute myeloid patients treated by intensive chemotherapy but not hypomethylating agents.
The authors addressed effectively referees’ comments and the revised draft is largely improved. Given the current available literature and on the same line of the comment raised by referee 3, I consider mis-leading the definition of LSCs for CD34+CD38-CD123+ bone marrow cells from AML patients. LSCs in AML have not been universally identified and it is highly likely that LSCs are quite heterogeneous and the phenotypic identification could be including different markers combinations depending on the AML subtype. Therefore, I would suggest to not indicate the CD34+CD38-CD123+ population as LSCs.
Author Response
We thank the reviewer for the comments. We acknowledge the term of LSC as inapropriate. We modified the manuscript taking it into account (see response to the academic Editor).